# African Swine Fever Vaccine Candidate ASFV-G-ΔI177L/ΔLVR Protects Against Homologous Virulent Challenge and Exhibits Long-Term Maintenance of Antibodies

**DOI:** 10.3390/ani15040473

**Published:** 2025-02-07

**Authors:** Sun A Choi, Yeonji Kim, Su Jin Lee, Seong Cheol Moon, Keun Seung Ahn, Xinghua Zheng, Do Soon Kim, Se Young Lee, Seung Pyo Shin, Dongseob Tark, Wonjun Kim, Yongwoo Shin, Weonhwa Jheong, Jung Hyang Sur

**Affiliations:** 1Central Research and Development Institute, Komipharm International Co., Ltd., Siheung-si 15094, Republic of Korea; dktjschl@komipharm.com (S.A.C.); leesoot@komipharm.com (S.J.L.); moonvet@komipharm.com (S.C.M.); pig182@komipharm.com (K.S.A.); xinghua@komipharm.com (X.Z.); kimds202345@komipharm.com (D.S.K.); man32@komipharm.com (S.Y.L.); sinseng1234@komipharm.com (S.P.S.); 2Wildlife Disease Response Team, National Institute of Wildlife Disease Control and Prevention (NIWDC), Ministry of Environment, Gwangju 62407, Republic of Korea; yeonchi16@korea.kr (Y.K.); kwj2208@korea.kr (W.K.); jams80@korea.kr (Y.S.); 3College of Veterinary Medicine, Chungnam National University, Daejeon 34134, Republic of Korea; 4Laboratory for Infectious Disease Prevention, College of Veterinary Medicine, Korea Zoonosis Research Institute, Jeonbuk National University, Iksan 54531, Republic of Korea; tarkds@jbnu.ac.kr

**Keywords:** African swine fever, domestic pigs, live attenuated vaccine, ASFV-G-ΔI177L/ΔLVR, safety, efficacy

## Abstract

African swine fever is a major threat to the global swine industry. The disease was once confined to specific geographical regions. However, it is now spreading rapidly across continents; this has raised concerns among researchers and pig farmers alike. Therefore, the development of a vaccine against African swine fever virus has recently become a top priority in veterinary research. In this study, the first group was challenged after vaccination, while the second group received the vaccine alone. The former group confirmed the efficacy and safety of the vaccine, while the later group’s high antibody levels remained stable for about two months. Our findings can help predict the defense potential of the vaccine against highly virulent field strains of African swine fever virus.

## 1. Introduction

African swine fever (ASF) is a deadly disease that affects both farm-raised and wild boars. ASF has spread from Africa through Europe to most countries in Southeast and North Asia, where it is endemic [1,2]. ASF outbreaks have caused considerable economic losses in most Asian countries since they were first reported in China in 2018 [3]. This disease manifests in various forms, ranging from very severe to subclinical infections depending on various factors [4,5]. ASF is caused by the African swine fever virus (ASFV), which is classified under the family *Asfarviridae* and genus *Asfivirus*. ASFV has a double-strand DNA genome that is 170–190 kbp in length, mainly owing to deletions and insertions in the left and right terminal genomic regions [6,7,8].

Effective prevention or treatment strategies are yet to be developed for the 25 ASFV genotypes; standard management of the disease in most countries is strict herd culling. ASF is a serious threat to the global swine industry, encompassing domestic, feral, and wild boar [9,10,11]. The disease is more than 100 years old. Although initial outbreaks were primarily confined to the African continent, the frequency of ASF outbreaks has increased since 2007 [12]. The disease has become an ongoing threat to the pork industry and global food security, consequently affecting prices, production, and international trade. Pork accounts for 36% of global meat consumption, making it the second most consumed protein source. Thus, eradicating ASF is a critical priority for ensuring sustainable pig production [13].

Attempts to produce live attenuated vaccines (LAVs) are underway. Various ASF vaccine candidates, including inactivated vaccines, recombinant subunit vaccines, vector vaccines, and LAV strains, have been developed in the last few decades [14,15,16,17,18]. However, vaccine safety remains largely understudied. Clinical trials of LAVs have consistently demonstrated protection against ASFV. Thus, LAVs are considered the most effective method for protection against ASFV infection to date [19,20,21,22,23,24].

ASFV-G-ΔI177L, an attenuated live vaccine candidate obtained by deleting the *I177L* gene from the genome of ASFV-G, was recently developed [19]. The safety of this vaccine candidate has been shown even at high-dose parenteral inoculation. It can effectively induce defense against the highly virulent parental strain ASFV-G, even at relatively low doses. However, this attenuated virus replicates only in primary cultures of porcine macrophages, which poses a challenge for commercial mass production. Therefore, ASFV-G-ΔI177L/ΔLVR, a new vaccine candidate, has been developed. This vaccine strain is derived from ASFV-G-ΔI177L with deletion of the left variable region (LVR) that replicates efficiently in Plum Island porcine epithelial cells (PIPECs) [20].

The safety, immunogenicity, and efficacy of ASFV-G-ΔI177L/ΔLVR is considered equivalent to that of ASFV-G-ΔI177L. However, ASFV-G-ΔI177L/ΔLVR is the first highly efficient ASF vaccine candidate that has been rationally designed to replicate in an existing cell line and is suitable for commercial vaccine production [20].

In this study, we aimed to evaluate the efficacy, safety, and long-term antibody persistence of the ASFV-G-ΔI177L/ΔLVR vaccine. Our findings confirm that the ASFV-G-ΔI177L/ΔLVR vaccine candidate can play a role in the prevention of ASF infection in the swine industry.

## 2. Materials and Methods

### 2.1. Cell Culture and Virus

The production of the Mast Seed Virus (MSV) based on cell culture requires ease of cultivation, genetic stability during passaging, and high viral yield. Thus, adherence to the quality standards outlined by the World Organization for Animal Health (WOAH) [25] is crucial. The biological properties of MSV were characterized as follows: The vaccine candidate strain, ASFV-G-ΔI177L/ΔLVR, was initially provided by the United States Department of Agriculture (USDA) as an eighth (P8) passage ASF live attenuated vaccine (ASF-LAV) cultured in PIPEC. A brief description of the PIPEC passages using a vaccine strain provided by the USDA is as follows: PIPECs are a subclone of the porcine fetal kidney cell line that stably expresses bovine αVβ6 integrin (20). Briefly, PIPEC cultures were maintained in Dulbecco’s Modified Eagle Medium (Gibco, Grand Island, NY, USA) supplemented with 10% fetal bovine serum (FBS; HyClone Laboratories, Logan, UT, USA), 1% penicillin-streptomycin (Gibco, Grand Island, NY, USA), and 2 mM L-glutamine. Cells were cultured at 37 °C in a humidified incubator with 5% CO_2_. Cell passages were conducted every 7–10 days; then, we read the mCherry expression using fluorescence microscopy [20]. After the first introduction, another passage was added to produce the ninth passage (P9). Subsequently, the cells underwent 10 further passages. The one that exhibited the highest vaccine titer was MSV (ASFV-G-ΔI177L/ΔLVR-P19), as confirmed by various genetic modifications.

Virus titration was conducted using primary swine macrophage cultures seeded in 96-well plates. Serial dilutions of the virus were prepared and inoculated onto the macrophage cultures, maintained in macrophage-specific medium. Viral presence was assessed through hemadsorption (HA) assay, with erythrocyte attachment to infected macrophages serving as the indicator of viral replication. Virus titers were calculated using the Reed and Muench method to determine the 50% hemadsorption dose [26].

Domestic pigs were immunized with the ASFV-G-ΔI177L/ΔLVR vaccine candidate. A highly virulent Korea/wildboar/ASFV-Hwacheon/2020 strain (GenBank accession number: OR159219.1) was used in for animal challenges. This field-isolated strain was provided by the Ministry of Environment in the Republic of Korea.

### 2.2. Animal Experiments

Animal experiments were conducted under animal biosafety level 3 (ABSL-3) conditions at the Korea Institute of Zoonotic Diseases. All experimental protocols were approved by the Institutional Animal Care and Use Committee (protocol JBNUNON 01-022-2023 and JBNUNON 2024-049). The experimental animals were 7–8-week-old commercial breed pigs. The ASFV-G-ΔI177L/ΔLVR vaccine candidate was used after confirmation of pre-vaccination serologic negativity for Porcine Circovirus 2, Mycoplasma hyopneumoniae, and Porcine Reproductive and Respiratory Syndrome Virus, which are major swine diseases that could affect the experiment. The animals were divided into two groups (Groups A and B) for intramuscular immunization.

Group A (1–4) was intramuscularly inoculated with 1 mL of ASFV-G-ΔI177L/ΔLVR with 10^2.25^ TCID_50_. Next, the pigs were challenged with 10^2^ HAD_50_ of a highly virulent Korean ASFV field strain Hwacheon/2020 intramuscularly 28 days post-vaccination (dpv). Samples including whole blood, rectal and oral swabs were then collected at 0, 4, 7, 10, 14, 18, 21, 25, and 28 dpv and at 4, 7, 11, and 14 days post-challenge (dpc) for vaccine genome copy and antibody identification, respectively. Group A-1 (1–4) consisted of four pigs and was intramuscularly challenged with 1 mL of 10^2^HAD_50_ of the Hwacheon/2020 strain. The non-vaccinated control group (A-2; 1–4) consisted of four animals raised without extra measures (Table 1). The blood, rectal, and oral swab sample collection schedule was the same as that for Group A. Clinical signs, rectal temperature, and survival rates were recorded daily. Animals that died or survived from vaccination to challenge (14 dpc) and were subsequently euthanized were necropsied for pathological analysis.

Group B (1–4) was vaccinated with the ASFV-G-ΔI177L/ΔLVR vaccine alone and subsequently tested for persistence of vaccine genome copies and antibodies for up to 56 days to confirm the immunogenicity and safety of the vaccine. Group B was further divided into two subgroups (Table 1). First, four pigs were immunized with 10^3^ TCID_50_ of ASFV-G-ΔI177L/ΔLVR (1 mL) via intramuscular administration, and the safety of the vaccine was monitored for 56 days without challenge. Whole blood, rectal, and oral samples were collected at 0, 4, 7, 11, 14, 18, 21, 24, 28, 32, 35, 39, 42, 46, 49, 53, and 56 dpv to analyze the vaccine genome copy as well as antibodies. In Group B-1 (1–2), the non-vaccinated control group, two animals were bred until 56 days. Whole blood, rectal, and oral samples were collected at the same intervals as those for Group B. 

### 2.3. Quantitative Real Time PCR (qPCR) for Detection of the ASFV Genome

qPCR was used to detect the virus in whole blood, rectal, and oral samples at vaccination. The ASFV DNA was isolated using the RSC Whole Blood & RSC Tissue DNA Kit (Maxwell^®^; Promega, Madison, WI, USA) and equipment (Maxwell^®^ CSC 48 Instrument IVD, Promega) according to the manufacturer’s instructions. The methods of DNA purification from whole blood samples are similar. Proteinase K solutions (20 µL) were added to each incubation tube containing 300 µL of whole blood. Additionally, 200 µL of lysis buffer was added to each incubation tube. Each tube was vortexed for 10 s and incubated in the heating block (set to 56 °C) for 20 min. DNA purification was then performed according to the manufacturer’s instructions, followed by the addition of 50 µL of elution buffer to extract the DNA.

ASF-specific real-time PCR was performed using the Vet max™ African swine fever virus detection kit (Thermo Fisher Scientific, Waltham, MA, USA), which has been validated and certified by the WOAH for ASFV detection (the p72 target gene) [22]. Ct values < 45 denoted positivity.

### 2.4. Detection of Anti-ASFV Antibodies

Whole blood was obtained from ASFV-G-ΔI177L/ΔLVR-inoculated animals, and serum levels of ASFV antibodies were evaluated using the ID Screen^®^ ASFV competitive enzyme-linked immunosorbent assay (cELISA; IDvet, rue Louis Pasteur, Grabels, France). The serum was separated from the whole-blood samples, and cELISA was performed according to the manufacturer’s instructions. S/N% was calculated using the following formula: $$S/N\%=\frac{\mathrm{OD}\;\mathrm{samples}\;-\;\mathrm{OD}\;\mathrm{positive}\;\mathrm{control}}{\mathrm{OD}\;\mathrm{negative}\;\mathrm{control}\;-\;\mathrm{OD}\;\mathrm{positive}\;\mathrm{control}}\times 100$$

Serum samples with S/N% ≤ 40% were considered positive. In contrast, serum samples with S/N% ≥ 50% were considered doubtful and their analysis repeated.

## 3. Results

### 3.1. Efficacy of ASFV-G-ΔI177L/ΔLVR Against ASFV FieldStrain Hwacheon/2020

Group A (1–4) was intramuscularly immunized with a live attenuated ASFV-G-ΔI177L/ΔLVR vaccine at 10^2.25^ TCID_50_/1mL/dose to confirm the safety and efficacy of this vaccine. As 28 dpv, 1 mL of a highly virulent homologous ASFV field strain (Hwacheon/2020) at 10^2^ HAD_50_ was challenged by IM. No unusual temperature changes were observed in the animals at 28 dpv. A slight transient increase in body temperature (40.1–41.5 °C) was observed in three animals after the challenge, but all recovered (Figure 1) and no unusual clinical findings were noted before euthanasia.

Group A-1 (1–4): The positive control group developed persistent high temperatures and ASF-related symptoms starting at 3 dpc with the ASFV field strain. All animals died between 11 and 14 days after challenge.

Group A-2 (1–4): in contrast, non-vaccinated animals remained clinically normal, with no signs of disease throughout the observation period.

### 3.2. Evaluation of ASF Vaccine DNA (ASFV p72 Gene) in Challenged Pigs After ASFV-G-ΔI177L/ΔLVR Vaccination

Group A (1–4): To detect the ASF genome in whole blood after vaccination with ASFV-G-ΔI177L/ΔLVR, samples collected at 0, 4, 7, 10, 14, 18, 21, 25, and 28 dpv were tested through real-time PCR targeting the *p72* gene. At 4 dpv, 50% (2/4) of the ASFV genome was at low detection levels in whole blood (Ct = 41.10 ± 3.91). This genome became 100% detectable (Ct = 31.25 ± 9.69) in all animals from day 14 to the time of challenge. Whole blood was collected on days 4, 7, 11, and 14 after the challenge for ASF genome detection. A relatively constant viral genome was observed from 4 dpc to 14 dpc (at euthanasia) (Ct = 25.53 ± 6.46) (Table 2).

Group A-1 (1–4): No viral genome was detected by the time of challenge on day 28. However, high detection levels (Ct = 16.68 ± 0.26) were observed at 4–11 dpc. All animals in this group died. Group A-2 (1–4) was used as a negative control, and no ASFV genome was detected. 

### 3.3. Evaluation of ASF Vaccine Antibodies in Challenged Pigs After ASFV-G-ΔI177L/ΔLVR Vaccination

Antibody detection using cELISA was read as a positive reference at S/N% <40%. Each group was challenged at 28th dpv, and whole blood was collected to evaluate antibody response to the vaccine (Table 3).

Group A (1–4): Antibody positivity was detected at 10 dpv with ASFV-G-ΔI177L/ΔLVR (25%, 1/4). Low-level (S/N% = 33.85 ± 4.80) antibody positivity was confirmed in all vaccinated groups on day 14. Additionally, antibody positivity (S/N% = 21.75 ± 6.84) persisted at high levels on day 28. Furthermore, 100% of animals survived at 4–14 dpc, with high (S/N% = 10.03 ± 2.86) antibody levels persisting after 28 dpc.

In Group A-1 (1–4), the ASF-positive control group, antibody levels approached the positive threshold at 7 dpc. However, all animals died due to infection by day 11. In Group A-2 (1–4), the non-vaccinated group, no detectable ASF-specific antibodies were observed throughout the study.

### 3.4. Evaluation of ASF Vaccine DNA in Rectal and Oral Swabs from the Post-Vaccine Challenge Group and Vaccination-Only (Long-Term) Group

Group A (1–4): ASFV genome detection in rectal and oral samples was found at significantly lower levels compared to whole blood. It was first detected in the oral cavity of one of four pigs on day 7 of vaccination (7 dpv) and was detected in both the rectal and oral cavity of two of four pigs on day 18 (dpv). By day 28, the day of challenge, ASFV genome detection remained below 50% of the vaccinated group. Following the challenge, moderate levels of ASFV genome were consistently detected in all pigs up to euthanasia at 14 dpc (Appendix A).

Group A-1 (1–4): as expected for a challenge control group, high levels of the ASFV genome were detected in the rectum and oral cavity at 7 dpc post-euthanasia (Appendix A).

Group A-2 (1–4): in contrast to the negative control group, all were confirmed negative (Appendix A).

Group B (1–4): On day 7 of vaccination (7 dpv), two of four pigs were initially identified with low levels of the ASFV genome in the rectum and oral cavity. After being detected in three out of four pigs on days 14–18 (14–18 dpv), only two out of four pigs were found to have very low levels of the ASFV genome until day 56 (56 dpv) when they were euthanized. According to Ct value, among rectal and oral samples, the ASFV genome was detected more frequently in oral samples (Appendix A). All Group B-1 (1–2) controls were ASFV-genome-negative. 

### 3.5. Pathological Findings in Pigs Challenged After ASFV-G-ΔI177L/ΔLVR Vaccination

Pigs that died or were euthanized in all ASF vaccination groups, including positive controls, were necropsied to evaluate typical ASF lesions. Normal findings were observed in major organs, including most lymph nodes (submandibular, inguinal, and gastrointestinal–hepatic lymph nodes), the spleen, kidney, and lungs, in pigs challenged after ASFV-G-ΔI177L/ΔLVR vaccination (Figure 2 and Figure 3, V1–V4). Histopathological lesions also showed no unusual findings (Figure 2 and Figure 3, V1-1 to V4-1). In contrast, positive controls challenged with strain Hwacheon/2020 had typical ASF lesions in most lymph nodes, the spleen, kidneys, and other organs. The most prominent macroscopic lesions were observed in the lymphoid system (Figure 2, C1–C4).

The spleen exhibited significant enlargement with rounded edges. It was characterized by a dark red appearance and mild splenomegaly (Figure 3, C1–C4). Lymph nodes were frequently swollen, edematous, and displayed hemorrhages in both the cortex and medulla. In the kidneys, severe hemorrhages were evident in the renal medullary region, including characteristic petechial hemorrhages on the surface. Histopathological examination of the lymph nodes revealed diffuse, moderate to severe hyperemia, with medullary sinuses infiltrated by mononuclear cells, pyknotic cells undergoing karyorrhexis, and cellular debris (Figure 2, C1-1 to C4-1). Similar findings were observed in the spleen, where the red pulp showed severe, diffuse hemorrhagic congestion filled with erythrocytes and pyknotic cells (Figure 3, C1-1 to C4-1). 

### 3.6. Long-Term Safety Evaluation After ASFV-G-ΔI177L/ΔLVR Vaccination in Growing Pigs

Group B (1–4) was administered the live attenuated vaccine ASFV-G-ΔI177L/ΔLVR (10^3^ TCID_50_/1mL) after 3-day stabilization following pig introduction to confirm the safety of the vaccine. However, we verified vaccine safety for 56 dpv and could not verify it further due to ABSL-3 usage restrictions.

Transient fever (40.0–41.3 °C) was observed during the first 3-day stabilization period in group B. Thereafter, fever and clinical symptoms were not observed by the time of euthanasia (Figure 4). As expected, non-vaccinated naive animals (Group B-1 (1–2)) remained clinically normal throughout the observation period.

### 3.7. Long-Term Evaluation of ASF Vaccine DNA (ASFV p72 Gene After ASFV-G-ΔI177L/ΔLVR Vaccination

Group B (1–4): Whole-blood samples collected at 0, 4, 7, 11, 14, 18, 21, 24, 28, 32, 35, 39, 42, 46, 49, 53, and 56 dpv were tested via real-time PCR targeting the p72 gene to detect the ASF genome after vaccination with ASFV-G-ΔI177L/ΔLVR. At 4 dpv, 75% (3/4) of the animals had moderate levels of the ASFV genome (Ct = 35.93 ± 6.61). The ASFV genome was detected in all animals by day 56, the time of euthanasia. At 28 dpv (Ct = 24.68 ± 2.32), the ASF genome was detected at a relatively high level. However, the ASFV genome levels progressively decreased until 46 dpv, with even lower levels identified at 56 dpv (Ct = 31.18 ± 1.96) (Table 4). All non-vaccinated Group B-1 (1–2) controls were ASFV-genome-negative.

### 3.8. Long-Term Evaluation of ASFV-G-ΔI177L/ΔLVR Vaccine Antibodies After Immunization

Group B (1–4): Antibody positivity was detected on day 11 of ASFV-G-ΔI177L/ΔLVR vaccination (50%, 2/4). Moderate (S/N% = 19.05 ± 4.76) and high (S/N% = 12.33 ± 5.22) levels of antibody positivity were detected in all vaccinated groups on days 24 and 28, respectively. ASF antibody levels were higher (S/N% = 7.23 ± 2.25) at 42 dpv. These levels were even higher (S/N% = 5.38 ± 2.24) at 56 dpv, approximately 2 months after ASF vaccination. All unvaccinated Group B-1 (1–2) controls were ASF-antibody-negative (Table 5).

### 3.9. Long-Term Pathological Evaluation After ASFV-G-ΔI177L/ΔLVR Vaccination

Group B: Major organs, including most lymph nodes (submandibular, inguinal, gastrointestinal–hepatic lymph nodes) and the spleen, were normal in pigs vaccinated with ASFV-G-ΔI177L/ΔLVR (Figure 5, V1–V4). Similarly, no macroscopic lesions were observed in the lymph nodes and spleen of non-vaccinated naive animals at necropsy (Figure 5, C1–C2). In addition, no histopathologic lesions were observed in most major organs of both vaccinated and non-vaccinated animals.

## 4. Discussion

The saying “Rome was not built in a day” rings true for research efforts towards ASFV eradication. ASFV is a deadly swine disease that was first reported in Africa in 1921 [27]. It reached continental Europe in 1957 and has since spread to Eastern Europe and then Asia. The global spread of this disease is devastating for the swine industry [16]. Thus, various forms of candidate vaccines have been developed. Furthermore, clinical trials have been ongoing for decades. Twenty-five ASFV genotypes have been identified to date, along with a lack of cross-reactivity [10].

Genetically engineered live attenuated vaccines have been developed to address the ASF pandemic. These include deletion mutant vaccines produced through homologous recombination. These promising ASF vaccine candidates are under evaluation in ongoing clinical trials. One of the first reported vaccine viruses to exhibit reasonable attenuation was “ASFV-G-∆9GL” [28], as reported in the ASF live attenuated vaccine (ASF-LAV) development status report. This virus possesses reduced residual virulence with additional deletion of the *UK* gene [29]. “BA71∆CD2”, another promising vaccine candidate, induces strong humoral and cellular responses and has shown promise for cross-protection against homologous viruses [30].

HLJ/18-7GD, another promising vaccine candidate, was fully attenuated in pigs. This vaccine strain is characterized by the deletion of seven genes. It has a low risk of transformation into a virulent strain and can induce robust protection against lethal ASFV challenge in pigs [31]. Most notably, “ASFV-G-∆I177L” [19] and “ASFV-G-∆MGF” [32] provide close-to-sterile protection against homologous challenge while exhibiting low vaccine-induced viremia. Although these strains were recently commercialized, they have been subject to a vaccine withdrawal notice from the USDA, owing to confirmed pathogenic reversion in post-vaccination animal-to-animal serogroups [33,34].

Of the many ASF-LAVs developed to date, only a few clinical trials have been reported for ASFV-G-ΔI177L/ΔLVR. Specifically, the rational development of ASFV-G-ΔI177L/ΔLVR facilitated that of ASFV-G-∆I177L, an attenuated live vaccine candidate obtained by deleting the *I177L* gene from the ASFV-G genome [20]. ASFV-G-ΔI177L is considered safe even at high-dose parenteral inoculation. In addition, it is highly effective against the highly virulent parental strain ASFV-G, even at relatively low doses. However, this attenuated virus replicates only in primary cultures of porcine macrophages, which poses a challenge for commercial mass production. This vaccine candidate is a derivative strain of ASFV-G-∆I177L with a deleted LVR that replicates efficiently in the PIPEC porcine cell line. Furthermore, ASFV-G-ΔI177L/ΔLVR is the first highly efficient ASF vaccine candidate rationally designed to replicate in an existing cell line. Moreover, it is considered suitable for commercial production [20]. 

We sought to investigate the efficacy and safety of the ASFV-G-ΔI177L/ΔLVR vaccine, which has already been confirmed in pregnant sows (submitted for publication). In the present study, we further evaluated the safety and efficacy of this vaccine in two independent animal experiments: Group A, pigs challenged with a homologous virulent ASFV field strain after vaccination to evaluate vaccine safety and efficacy, and Group B, domestic pigs reared for 56 days after vaccination for vaccine genome detection and antibody persistence analysis. 

In Group A, complete protection was confirmed in vaccinated pigs, and the ASF genome copy (*p72*) was detected in the whole blood of two out of four pigs from the 2nd day of immunization. The detection was confirmed at a constant level (Ct = 2.53 ± 6.46) from 14 to 28 dpc. Moreover, ASF *p72* detection in the vaccine group was more than 45% lower than that in the control group. Fewer vaccine genome copies over time may eventually lead to viral clearance. From the perspective of ASF vaccine virus shedding, the ASFV genome was detected at very low levels in rectal and oral swabs compared to whole blood up to the pre-challenge time point (28 dpv), with inconsistent detection among individual animals. Overall, the frequency of ASFV genome detection was higher in the oral cavity than in the rectal cavity. However, following the challenge, the ASFV genome was consistently detected at low to moderate levels in both rectal and oral swabs across all animals.

Autopsy findings revealed no ASF lesions in the organs of pigs challenged after vaccination. Similarly, no specific histopathological lesions were observed in most organs. However, a key concern is the assessment of vaccine virus shedding in vaccinated animals and the shedding of the challenge virus post-vaccination. Virus shedding in vaccinated animals is a critical factor influencing both safety and efficacy. Detection of the ASF *p72* genome associated with the vaccine virus for a specific duration post-vaccination may inhibit viral replication and promote the development of sterile immunity against the homologous virus [16,19]. We acknowledge that no live attenuated vaccine can be considered entirely perfect. Viral clearance following vaccination refers to the process by which the immune system, stimulated by the vaccine, eliminates the virus from the body. An effective vaccine should induce a robust immune response capable of significantly reducing and ultimately eradicating the virus. However, the rate and efficiency of viral clearance are influenced by several factors, including the type of vaccine, the health status of the host, and the characteristics of the specific virus. In the case of ASF, ongoing research continues to explore the impact of vaccination on reducing viral presence and shedding, highlighting the complexity of achieving effective viral clearance in vaccinated animals.

Animals with no detectable vaccine genome copies with a Ct < 45 for a certain period after vaccination exhibited no effective defense against the challenge virus and consequently died. These results are consistent with those reported in previous studies [16,19]. In addition, the detection of consistently high levels of antibodies after ASFV-G-ΔI177L/ΔLVR vaccination is particularly important for assessing immunoprotective effectiveness. In the present study, all vaccinated animals showed excellent antibody positivity that consistently increased over time. These results demonstrate that animals with high vaccine ASF *p72* genome and antibody levels after vaccination have 100% survival.

Group B was part of a clinical trial for long-term breeding of domestic pigs after ASFV-G-ΔI177L/ΔLVR vaccination to detect the vaccine genome and antibody persistence. Low levels of the ASF p72 genome were detected in three out of four pigs on day 4 of vaccination, and over time, more genome copies could be detected on day 28 of vaccination. However, a gradual decline in the number of genome copies was observed around 42 days after vaccination. In contrast, vaccine antibodies were detected in two out of four pigs on day 11 with S/N% < 40, the criterion for antibody positivity, and a relatively high level of vaccine antibodies (S/N% = 19.05 ± 4.76) was detected from day 24. High levels of antibodies were maintained at day 42 (S/N% = 7.23 ± 2.25), more than one month after vaccination, and even higher ASF vaccine antibodies (S/N% = 5.38 ± 2.24) were observed at day 56, approximately two months post vaccination. The persistence of high ASF vaccine antibodies in the present study suggests that exposure to a highly virulent field ASFV strain provides adequate protection. In the long-term ASF-vaccinated group, vaccine virus shedding was detected at low levels in the rectum and oral cavity in three out of four pigs between days 14 and 18 post-vaccination. Subsequently, very low levels of viral shedding were observed in the oral cavity (Ct = 42.7 ± 2.3) in only two pigs on day 56, at the time of euthanasia. These findings indicate that vaccine virus shedding was minimal and significantly limited approximately two months after administration of the ASF-G-ΔI177L/ΔLVR vaccine. However, we could not use more animals for the experiment, and all were euthanized on day 56, due to ABSL-3 space and time limitations.

ASF prevention is a major challenge for the global pig-rearing industry. Scientific advancements in the last few decades have enabled the progression of various vaccine development strategies focusing on subunit vaccines, inactivated vaccines, multiple gene-deleted vaccines (live attenuated vaccines), and genetic and natural attenuated vaccines. The most field-ready vaccine candidate to date is a live attenuated vaccine. Although effective preventive or therapeutic strategies for 25 ASFV genotypes are still lacking, considerable progress has been made in the protection of animals against Genotype II, the most prevalent genotype in each continent [35]. While clinical trials have demonstrated the efficacy of ASF live attenuated vaccines, limited results have been reported for the ASFV-G-ΔI177L/ΔLVR vaccine candidate. 

Although this study had some limitations, we believe that the ASFV-G-ΔI177L/ΔLVR vaccine provides at least as much protection against ASFV as any live attenuated vaccine candidate developed to date. Future challenges will include vaccinating sufficient domestic pigs and ensuring that vaccine antibody levels persist through 5 to 6 months of average market age. In addition, future studies are required to identify and challenge the inflection point for vaccine antibodies after the first dose. Finally, sufficient analysis of ASF target tissues (target organs, whole blood, oronasal swab, etc.) is required to demonstrate additional sterile immunity.

## 5. Conclusions

For decades, ASF live attenuated vaccines have been developed through various genetic modifications, with numerous clinical trials conducted. However, clinical trials of ASFV-G-ΔI177L/ΔLVR vaccine candidates have been markedly limited. In this study, we strictly followed the WOAH Biological Standard Commission vaccine production standards and conducted clinical trials with a high-purity vaccine. Vaccine purity was achieved by thoroughly eliminating the possibility of mutations during cell line passage. In this study, we report excellent efficacy and safety in homologous challenge after low-dose ASFV-G-ΔI177L/ΔLVR vaccination. In addition, ASF vaccine antibodies were maintained at a substantially high level for approximately 2 months after vaccination. These results confirm that the ASFV-G-ΔI177L/ΔLVR vaccine candidate can play a role as a relief pitcher for ASF infection in the swine industry. 

## Figures and Tables

**Figure 1 animals-15-00473-f001:**
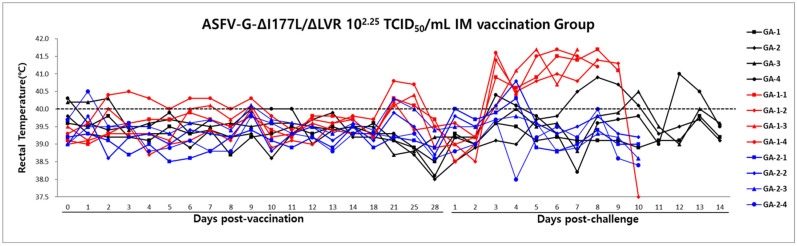
Rectal temperature and survival of the pigs following vaccination and challenge. The dotted line indicates the threshold for fever at 40.0 °C. The black line represents the challenge at 28 dpv; the red and blue lines indicate the challenge infection and negative control, respectively.

**Figure 2 animals-15-00473-f002:**
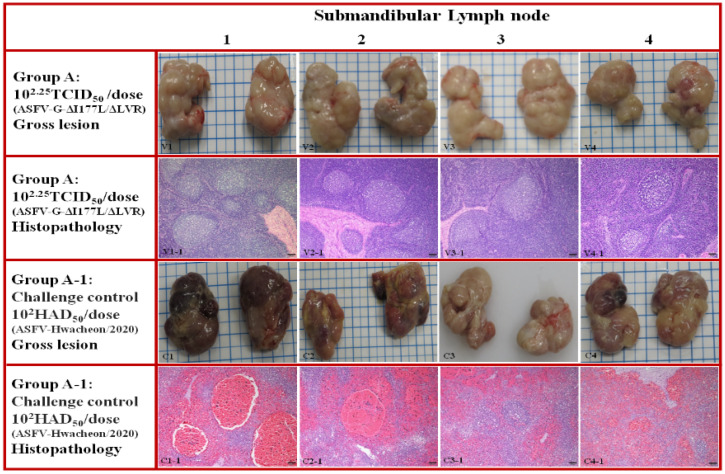
Group A: the submandibular lymph node from a pig vaccinated with ASFV-G-ΔI177L/ΔLVR before challenging. Macroscopic lesions (V1 to V4): all submandibular lymph nodes were normal. Microscopic lesions (V1-1 to V4-1): Sections of the submandibular lymph node. H&E stain, 100x Normal histopathological findings were confirmed in all pigs challenged after vaccination. Group A-1: submandibular lymph node from a pig challenged with the highly virulent ASFV isolate (Hwacheon/2020). Macroscopic lesions (C1 to C4): lymph nodes were swollen and edematous, with severe hemorrhages in both the cortex and medulla. Microscopic lesions (C1-1 to C4-1): Sections of the submandibular lymph node. H&E stain, 100x. Scale bar = 100 µm. Severely lymphocytic depletion affects both lymphoid follicle and is accompanied by severe hemorrhages in the medullary cords, blood infiltration of the marginal zones, and congestion of numerous vessels.

**Figure 3 animals-15-00473-f003:**
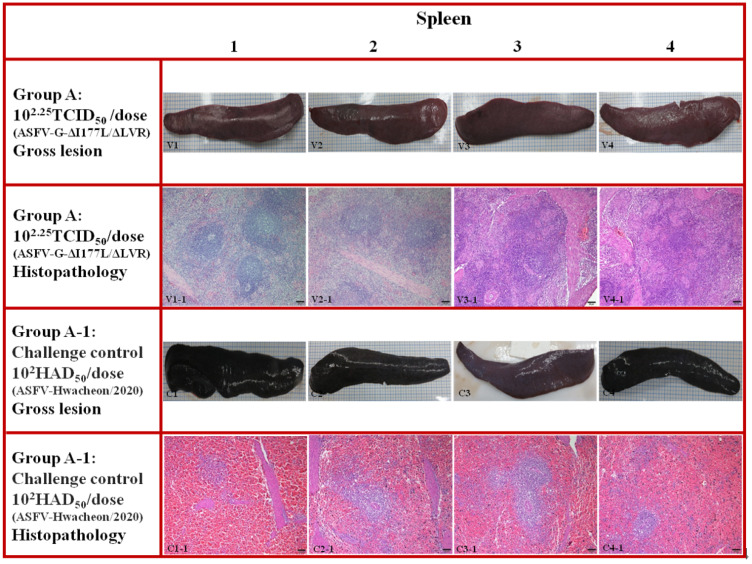
Group A: the spleen from a pig vaccinated with ASFV-G-ΔI177L/ΔLVR before viral challenge. Macroscopic lesions (V1 to V4): all spleens were normal in size and red in color. Microscopic lesions (V1-1 to V4-1): Sections of the spleen. H&E stain, 100x Normal histopathological findings were confirmed in all pigs challenged after vaccination. Group A-1: the spleen from a pig challenged with strain Hwacheon/2020. Macroscopic lesions (C1 to C4): the spleen shows an increase in size with round edges, along with friable (C1) and dark red to black color (C1 to C4). Microscopic lesions: Sections of the spleen (C1-1 to C4-1). H&E stain, 100x. Scale bar = 100 µm. Diffuse infiltration of the pulp by erythrocytes and macrophages showing severe lymphocytic depletion along with abundant pyknotic cells.

**Figure 4 animals-15-00473-f004:**
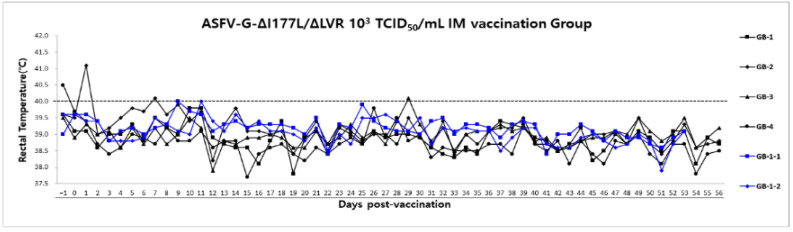
Rectal temperatures and survival of the pigs following vaccination. The dotted line indicates the threshold for fever at 40.0 °C. The black line represents monitoring up to 56 days after the first vaccination: The blue line indicates negative control.

**Figure 5 animals-15-00473-f005:**
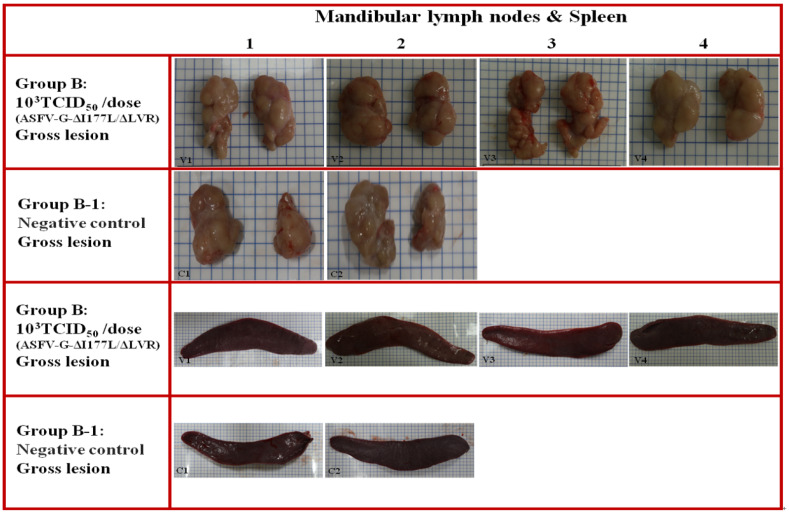
Group B: the submandibular lymph node from a pig vaccinated with ASFV-G-ΔI177L/ΔLVR. Macroscopic lesions (V1 to V4): the submandibular lymph node of long-term-vaccinated pigs was normal. Group B-1: the submandibular lymph node from a non-vaccinated pig (negative control). Macroscopic lesions (C1 to C2): all submandibular lymph nodes were normal. Group B: the spleen from a pig vaccinated with ASFV-G-ΔI177L/ΔLVR. Macroscopic lesions (V1 to V4): the spleens of long-term-vaccinated pigs were normal, exhibiting a bright red color. Group B-1: the spleen from a non-vaccinated pig (negative control). Macroscopic lesions (C1 to C2): all spleens were normal.

**Table 1 animals-15-00473-t001:** Summary of the ASFV vaccine clinical trial design.

Trial Group	Animal No. (No. of Pigs)	Vaccine/Dose	Challenge/Dose	Route
A	4 (1–4)	ASFV-G-ΔI177L/ΔLVR/10^2.25^ TCID_50_	ASFV-Hwacheon/2020(Genotype-II)/10^2.0^ HAD_50_	IM
A-1	4 (1–4)	Not applicable	ASFV-Hwacheon/2020(Genotype-II)/10^2.0^ HAD_50_	IM
A-2	4 (1–4)	Not applicable	Not applicable	-
B	4 (1–4)	ASFV-G-ΔI177L/ΔLVR/10^3.0^ TCID_50_	Not applicable	IM
B-1	2 (1–2)	Not applicable	Not applicable	-

IM: Intramuscular immunization.

**Table 2 animals-15-00473-t002:** Viral genome copies (* Ct value) in blood samples of pigs intramuscularly inoculated with ASFV-G-ΔI177L/ΔLVR before challenge with virulent ASFV-Hwacheon/2020.

Group		Ct * (qPCR, Whole Blood)													
		DPV (Days Post-Vaccination)							DPC (Days Post-Challenge)						
		0	4	7	10	14	18	21		25	28	4	7	11	14
10^2.25^ TCID_50_ /dose (IM)	GA1	45.00	45.00	37.6	36.6	35.7	33.6	39.0		36.3	30.4	30.5	33.1	34.9	36.7
	GA2	45.00	45.00	45.00	45.00	33.6	25.1	25.8		26.9	30.8	36.9	18.5	19.8	21.2
	GA3	45.00	36.9	20.8	19.5	21.5	21.2	21.6		21.9	24.3	23.4	23.7	21.1	22.0
	GA4	45.00	37.5	26.4	19.7	22.8	23.0	22.2		23.2	27.7	25.8	26.9	25.9	22.2
Challenge Control	GA-1-1	45.00	45.00	45.00	45.00	45.00	45.00	45.00		45.00	45.00	16.6	13.5	D **	D
	GA-1-2	45.00	45.00	45.00	45.00	45.00	45.00	45.00		45.00	45.00	16.6	15.3	D	D
	GA-1-3	45.00	45.00	45.00	45.00	45.00	45.00	45.00		45.00	45.00	17.1	14.1	D	D
	GA-1-4	45.00	45.00	45.00	45.00	45.00	45.00	45.00		45.00	45.00	16.4	14.1	D	D
Negative Control	GA-2-1	45.00	45.00	45.00	45.00	45.00	45.00	45.00		45.00	45.00	45.00	45.00	45.00	45.00
	GA-2-2	45.00	45.00	45.00	45.00	45.00	45.00	45.00		45.00	45.00	45.00	45.00	45.00	45.00
	GA-2-3	45.00	45.00	45.00	45.00	45.00	45.00	45.00		45.00	45.00	45.00	45.00	45.00	45.00
	GA-2-4	45.00	45.00	45.00	45.00	45.00	45.00	45.00		45.00	45.00	45.00	45.00	45.00	45.00

* Ct < 45 was considered positive. ** D: death.

**Table 3 animals-15-00473-t003:** Antibodies in blood samples of pigs intramuscularly inoculated with ASFV-G-ΔI177L/ΔLVR before and after challenge with virulent ASFV-Hwacheon/2020.

Group		S/N% * (cELISA)													
		DPV (Days Post-Vaccination)							DPC (Days Post-Challenge)						
		0	4	7	10	14	18	21		25	28	4	7	11	14
10^2.25^ TCID_50_ /dose	GA1	74.2	79.1	69.4	48.9	38.2	36.5	33.0		24.0	20.0	14.3	12.2	11.9	8.8
	GA2	77.5	82.3	58.0	39.6	39.0	39.3	38.2		36.6	26.4	31.5	13.5	8.4	9.3
	GA3	93.0	89.9	84.5	45.1	28.2	32.1	24.2		16.6	11.4	9.9	8.1	9.5	7.2
	GA4	78.2	79.4	76.6	64.3	30.0	33.1	30.0		31.1	29.2	21.7	17.5	20.6	14.8
Challenge Control	GA-1-1	77.8	86.5	86.8	80.7	82.3	76.6	78.2		68.6	76.3	81.5	49.9	D **	D
	GA-1-2	84.2	80.9	76.1	72.0	75.5	79.5	85.7		85.2	87.3	86.9	50.4	D	D
	GA-1-3	89.9	94.5	86.8	87.1	90.0	86.0	91.7		88.1	76.5	68.8	49.7	D	D
	GA-1-4	85.0	79.2	78.5	76.7	82.0	72.5	73.2		72.0	72.4	79.6	44.7	D	D
Negative Control	GA-2-1	83.8	87.4	87.5	94.2	89.5	71.1	68.7		76.2	77.4	85.8	75.6	91.0	94.5
	GA-2-2	89.2	92.5	76.3	92.4	85.6	85.2	76.3		87.4	83.0	92.8	82.6	99.9	95.4
	GA-2-3	87.4	83.4	67.2	86.1	85.4	84.4	80.8		79.3	82.2	86.2	79.4	102.4	91.3
	GA-2-4	66.0	70.9	68.1	85.9	83.3	81.1	78.5		68.4	73.6	75.4	72.0	99.2	91.8

* S/N% values ≤ 40% were considered positive. ** D: death.

**Table 4 animals-15-00473-t004:** Viral genome copies (* Ct value) in blood samples of pigs intramuscularly inoculated with ASFV-G-ΔI177L/ΔLVR.

Group		Ct * (qPCR, Whole Blood)																
		DPV (Days Post-Vaccination)																
		0	4	7	11	14	18	21	24	28	32	35	39	42	46	49	53	56
10^3^ TCID_50_ /dose(IM)	GB1	45.00	29.92	19.50	21.58	21.85	22.36	23.14	26.61	26.81	27.58	29.22	31.28	31.47	30.56	30.31	32.73	31.63
	GB2	45.00	29.32	20.15	21.29	21.92	21.93	22.21	23.82	24.60	25.27	26.23	27.62	29.09	28.65	28.94	27.50	28.97
	GB3	45.00	45.00	38.30	27.68	28.94	32.34	34.06	36.54	26.38	23.53	25.18	28.54	30.64	30.48	30.67	32.89	34.15
	GB4	45.00	39.49	41.06	45.00	45.00	45.00	45.00	24.55	20.93	22.07	23.04	24.55	26.00	26.79	27.79	30.35	29.96
Negative Control	GB-1-1	45.00	45.00	45.00	45.00	45.00	45.00	45.00	45.00	45.00	45.00	45.00	45.00	45.00	45.00	45.00	45.00	45.00
	GB-1-2	45.00	45.00	45.00	45.00	45.00	45.00	45.00	45.00	45.00	45.00	45.00	45.00	45.00	45.00	45.00	45.00	45.00

* Ct values < 45 were considered positive.

**Table 5 animals-15-00473-t005:** Antibodies in blood samples of pigs intramuscularly inoculated with ASFV-G-ΔI177L/ΔLVR.

Group		S/N% * (cELISA)																
		DPV (Days Post-Vaccination)																
		0	4	7	11	14	18	21	24	28	32	35	39	42	46	49	53	56
10^3^ TCID_50_ /dose (IM)	GB1	86.0	82.0	71.2	21.2	24.2	23.0	21.8	19.7	18.4	16.0	13.9	12.6	9.7	10.8	9.6	10.1	9.2
	GB2	85.2	77.8	57.8	12.1	13.4	14.5	15.1	11.1	9.3	10.2	9.9	9.1	8.8	7.2	6.3	4.7	4.4
	GB3	76.2	78.4	56.9	40.1	45.3	44.4	33.0	22.3	16.2	10.9	8.3	7.8	6.5	6.0	4.8	4.7	4.4
	GB4	88.4	87.1	73.7	56.4	59.7	59.4	45.8	23.1	5.4	3.8	3.6	3.9	3.9	3.5	2.9	3.6	3.5
Negative Control	GB-1-1	86.6	83.0	87.1	83.4	84.9	87.7	87.1	83.5	69.2	70.0	78.2	76.7	73.7	75.4	74.9	-	-
	GB-1-2	79.6	80.8	83.0	87.5	89.3	88.6	90.7	86.6	78.7	83.9	82.8	80.3	77.7	81.0	85.1	-	-

* S/N% values ≤ 40% were considered positive.

## Data Availability

The data and questionnaires that support the findings of this study are available from the corresponding author upon reasonable request.

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
