# Peer review of "African Swine Fever Vaccine Candidate ASFV-G-ΔI177L/ΔLVR Protects Against Homologous Virulent Challenge and Exhibits Long-Term Maintenance of Antibodies"

_animals, 2025, doi:10.3390/ani15040473_

Round 1
Reviewer 1 Report
Comments and Suggestions for Authors
In this study, Choi et al. aimed to evaluate the efficacy, safety, and long-term persistence of antibodies elicited by the ASFV-G-ΔI177L/ΔLVR vaccine. The findings confirmed the vaccine's efficacy and safety, demonstrating that antibody levels remained stable for approximately two months. The safety and efficacy of the ASFV-G-ΔI177L/ΔLVR vaccine candidate strain, along with the induced antibody response, have already been investigated. Consequently, the significance of this research is somewhat constrained. Some issues need to be addressed by the authors.
1. The group A lacks a high-dose immune group to evaluate the safety of ASFV-G-ΔI177L/ΔLVR.
2. The safety of the live attenuated ASF vaccine should be assessed by monitoring virus shedding; however, this study did not perform these tests.
3. The contents of Table 1 require redesigning to enhance the clarity of the experimental design for the readers.
4. Why did all four pigs in Group A-2 die at 10 dpc? Please explain.
5. In Group B of the experiment, the observation period was insufficient; a minimum duration of six months is necessary for a scientifically rigorous analysis of the dynamic changes in ASFV antibodies.
Author Response
Reviewer 1:
Comments and Suggestions for Authors
Comments 1: In this study, Choi et al. aimed to evaluate the efficacy, safety, and long-term persistence of antibodies elicited by the ASFV-G-ΔI177L/ΔLVR vaccine. The findings confirmed the vaccine's efficacy and safety, demonstrating that antibody levels remained stable for approximately two months. The safety and efficacy of the ASFV-G-ΔI177L/ΔLVR vaccine candidate strain, along with the induced antibody response, have already been investigated. Consequently, the significance of this research is somewhat constrained. Some issues need to be addressed by the authors.
- The group A lacks a high-dose immune group to evaluate the safety of ASFV-G-ΔI177L/ΔLVR.
The author respond: Thank you for your key questions. Our clinical investigators did not include the results of high-dose vaccine inoculation in this paper for the following reasons: We have conducted more than 10 experiments under ABSL-3 conditions for all of our ASF vaccine trials. High-dose vaccine inoculation is not in this experiment, but in previous experiments, we have already verified the efficacy and safety of the vaccine (100% survived) using 5 animals in each group with ASFV-G-ΔI177L/ΔLVR, ASFV-G-ΔI177L/ΔLVR, 106TCID50/dose. However, as suggested by the reviewer, we have organized the results of vaccine genome copy (qPCR) and vaccine antibodies (cELISA) into Table 1 (Genome copy) and Table 2 (Vaccine antibody) for easier understanding.
Table 1. Vaccine genome copies after vaccination (ASFV-G-ΔI177L/ΔLVR).
Table 2. Vaccine antibody after vaccination.
Comments 2. The safety of the live attenuated ASF vaccine should be assessed by monitoring virus shedding; however, this study did not perform these tests.
The author responds: Thank you for pointing out this important aspect.
The reason why we did not include it in the current paper is that we focused on whole blood results and vaccine efficacy, and if we added data on oral and rectal, the volume of the paper would be too large as well as the content, so we could not attach it. The qPCR (Ct<45) results for Rectal and Oral swabs are attached as Table 1-2. Although not confirmed by marketing age, the results for rectal and oral attached to the reviewer's virus shedding question show that low level genome copies are confirmed in oral swabs in 2 out of 4 dogs on day 56 (dpv) when vaccinated alone (Table 2). After checking, we will decide whether to attach them to this paper according to the reviewer's decision.
Summary) Vaccine genome shedding was identified at low levels in 2 out of 4 animals per individual, with somewhat more frequent shedding in the oral than the rectal. In the post-vaccine challenge, shedding was relatively consistent across all groups up to the time of euthanasia.
Summary) Vaccine genome shedding was detectable in all but 1 pig from day 14 post-vaccination and was mostly detected in oral swabs by day 48 (dpv) but was detected at very low levels in the oral cavity of 2 of 4 pigs by day 56 (dpv).
Comments 3. The contents of Table 1 require redesigning to enhance the clarity of the experimental design for the readers.
The author responds: Thank you. We've made an adjustment to make it clearer as you suggested. Here is the newly corrected Table 1 as suggested (see Line 124).
Table 1. Summary of the ASFV vaccine clinical trial design.
Trial Group |
Animal No. (No. of pigs) |
Vaccine / Dose |
Challenge / Dose |
Route |
A |
4 (1-4) |
ASFV-G-ΔI177L/ΔLVR / 102.25 TCID50 |
ASFV-Hwacheon/2020(Genotype-II) / 102.0 HAD50 |
IM |
A-1 |
4 (1-4) |
Not applicable |
ASFV-Hwacheon/2020(Genotype-II) / 102.0 HAD50 |
IM |
A-2 |
4 (1-4) |
Not applicable |
Not applicable |
- |
B |
4 (1-4) |
ASFV-G-ΔI177L/ΔLVR / 103.0 TCID50 |
Not applicable |
IM |
B-1 |
2 (1-2) |
Not applicable |
Not applicable |
- |
*IM: Intramuscular immunization
Comments 4. Why did all four pigs in Group A-2 die at 10 dpc? Please explain.
The author responds: Thank you for your point. The positive control group was challenged with the ASFV virulent field strain. Most survived until day 10, but some individuals were close to death and were synchronized for euthanasia and necropsy on day 10.
Comments 5. In Group B of the experiment, the observation period was insufficient; a minimum duration of six months is necessary for a scientifically rigorous analysis of the dynamic changes in ASFV antibodies.
The author responds: Thank you for pointing this out as well. As you pointed out, a scientifically rigorous analysis of the dynamic changes in ASFV antibodies requires confirmation by market age. However, we only used the ABSL-3 facility for about 60 days due to the limited use of the ABSL-3 facility. This reason is already mentioned in the paper (Lines 397-398, 410-411).
Minor comments
- The authors have to use the appropriate terminology. ASF is a disease, and ASFV is a causative agent of the disease.
The author responds: Yes.
- Line 51: Please cite the accurate reference supporting that ASFV also causes subclinical infection.
The author responds: Yes. We have added a reference to ASFV also causes subclinical infection in the sentence as follows (see Line 49).
Life (Basel). 2022 Aug 17;12(8):1255. doi: 10.3390/life12081255
African Swine Fever Virus: A Review
- Line 98: Not P19.
The author responds: Yes. I will delete P19 from the sentence (see Line 93). However, P19 means that the vaccine strain has been passaged in PIPEC cells 10 more times since it was first obtained from the USDA.
- Line 116: Please describe the route of experimental challenge.
The author responds: Yes. The route of inoculation was written as follows: the pigs were challenged with 102HAD50 of a highly virulent Korean ASFV field strain Hwacheon/2020 by intramuscularly on day 28 days post-vaccination (dpv). (see Line 112)
- Line 148: The DNeasy is the extraction kit from Qiagen.
The author responds: Thank you. We have rechecked the whole blood DNA kit and confirmed that it is not produced by QIAGEN but by Promega (see Line 138-139). For reference, you can check it on the following website. Maxwell® RSC Whole Blood DNA Kit
Comments 6. Line 365: ASFV-365 G-ΔI177L/ΔLVR was developed by USDA ARS.
The author responds: I understand. We've removed this sentence (see Line 355).
Reviewer 2 Report
Comments and Suggestions for Authors
Major comments
1. Line 99: Please provide the genetic information of P10 passages.
2. Materials and methods: The experiment was performed with a limited number of animals. Please calculate the power of the experimental design.
3. Materials and methods: Please describe the methodology of virus titration for preparing the vaccine and challenge inoculum. Also, please provide the back-titer of the vaccine and challenge inoculum.
4. Results: Even though the vaccinated pigs with ASFV LAV remained clinically normal with no fever, it can greatly affect the growth rate of pigs and subsequently there could be a significant difference in size between vaccinated and control pigs. Did the authors weigh the pigs?
5. Lines 378: It is important to interpret the data carefully. The authors found the clinical and pathological protections in vaccinated pigs, however, failed to observe the complete virological protection in vaccinated pigs. Thus, it indicates that the vaccine does not provide complete protection or sterile immunity.
6. Lines 384-390: The authors are aware of the importance of virus shedding in vaccinated pigs. However, no data is presented in this study. Please include virus shedding data.
7. One of the key findings in this study is the persistent infection of the vaccine strain in pigs up to 56 DPV. It could provide the favorable environments for viruses to acquire genetic mutations. Thus, it is important to determine the viral sequences in vaccinated pigs. In particular, viremia in vaccinated pigs on the day of the challenge may result in the recombination event between the vaccine strain and the challenge strain because recombination contributes to the genetic expansion of ASFV and the reduction in the efficacy of the current vaccine in the field. Please perform the NGS on the samples from Group A and B to support the safety of vaccine.
Minor comments
1. The authors have to use the appropriate terminology. ASF is a disease, and ASFV is a causative agent of the disease.
2. Line 51: Please cite the accurate reference supporting that ASFV also causes subclinical infection.
3. Line 98: Not P19.
4. Line 116: Please describe the route of experimental challenge.
5. Line 148: The DNeasy is the extraction kit from Qiagen.
6. Line 365: ASFV-365 G-ΔI177L/ΔLVR was developed by USDA ARS.
Author Response
Reviewer 2:
Comments 1. line 99: Please provide the genetic information for the P10 line.
The author responds: The point in line 99 refers to P19, which is an additional 10 passages of the PIPEC cell line from USDA. We performed NGS (Next-Generation sequencing) analysis of P19. The NGS analysis of the ASFV vaccine candidate P19 was performed by a commercial (Celemics) organization, although the analysis is ongoing, initial analysis suggests that the absence of genetic sequence variation between the USDA-introduced P9 and the serially passaged P19.
Here is a brief explanation of the analysis method: Genomic DNA was fragmented as approximately 200bp use of Celemics enzymatic preparation kit (Celemics, Seoul, Republic of Korea) and processed for Illumina sequencing by following steps; adapter ligation and pre-PCR for indexed NGS library. To capture all target regions, NGS libraries and probes were hybridized using Celemics target capture Kit (Celemics, Korea). Customized capture probes were designed to target region and optimized for chemical hybridization. Swine DNA blocker was made by Celemics Inc. and used instead of human DNA blocker. After washing process captured DNA library was then further amplified by post-PCR to obtain enough amount of sample for sequencing. Pooled libraries containing captured DNA library were subsequently sequenced on Illumina NextSeq 500 Sequencing System as 2 x 151bp paired-end reads.
Comments 2. Materials and methods: The experiment was performed on a limited number of animals. Please calculate the power of the experimental design.
The author responds: Let me explain why we used a limited number of animals in this study. There are two ABSL-3 facilities in Korea. Both are government organizations. The ABSL-3 facility we used is a relatively small facility installed in a university (College of Veterinary Medicine, Korea Zoonosis Research institute). Although it is required to use more than 5 animals per group, it was not possible to breed more than 4 animals in the breeding room compartment space. In addition, we used the ABSL-3 facility for about 60 days due to the limited period of use. These reasons are already mentioned in the paper (Lines 397-398, 410-411).
Comments 3. Materials and methods: Please describe the viral titration method for preparing the vaccine and attenuated inoculum and provide the titers of the vaccine and attenuated inoculum.
The author responds: The vaccine titers used in this ASF clinical trial were as follows (It is also clearly indicated in Table1)
Group A (Vaccine Titer), After vaccination and challenged: 102.25 TCID50/dose,
Group B (Vaccine Titer), Vaccine only: 103.0 TCID50/dose
Challenged ASFV: 102.0 HAD50/dose
Method of Virus Titration:
The 1) Vaccine titration protocol for vaccination and 2) Virus Titration protocol for challenge are different.
1) Vaccine strain titration protocol:
Prepare vaccine strain using PIPEC cells
- 10-fold dilution of virus in Free FBS 1X DMEM media.
- dispense 100 μl of virus dilution into a 96-well plate.
iii. Add 2x105 PIPEC cells to the 96-well plate using proliferation media (1X DMEM+10% FBS+1% Sodium pyruvate+1% Anti-Anti+0.1% Gentamicin Reagent Solution).
Add 100 μl of PIPEC cells 2x105 cells/ml to each well. 4.
- Incubate at 37℃ CO2 incubator for 7~10 days, then read the mCherry expression using fluorescence microscopy.
2) Preparation of attack inoculum using PAM cells virus titration
- dispense 4~5x106 cells/ml of PAM cells into a 96-well plate and prepare to monolayer in a 37℃ CO2 incubator.
- take out the cultured cell plate and wash once with free FBS 1X RPMI1640 media.
iii. make a 10-fold dilution of virus in free FBS media.
- Dispense 100 μL of virus into a 96-well cell plate. 5.
- incubate in a CO2 Incubator for 1hr/37℃.
- take out the plate, discard the virus dilution and wash twice with free FBS 1X RPMI1640 media.
vii. Add 100 μl/well of proliferation media (1X RPMI1640+10% FBS+1% Sodium pyruvate+1% MEM NEAA+1% HEPES+1% Anti-Anti+1% P/S) with 1% RBCs.
Add 100 μl/well.
viii. Incubate at 37℃ CO2 incubator for 5-6 days, then check for HAD using light microscopy.
Comments 4. Results: Even though the vaccinated pigs with ASFV LAV remained clinically normal with no fever, it can greatly affect the growth rate of pigs and subsequently there could be a significant difference in size between vaccinated and control pigs. Did the authors weigh the pigs?
The author responds: The pigs used in this study were healthy 8-week-old pigs. The pigs were provided with an average of 18 kg once before the introduction, and their body weights were not measured during the experiment. This was because the portable scale could not be moved to another compartment in the ABSL-3 breeding room, which was designed with a limited space, so body weights could not be measured. However, since they were fed ad libitum, generally normal growth was confirmed in the normal control group, vaccine group, and challenge group.
Comments 5. Lines 378: It is important to interpret the data carefully. The authors found the clinical and pathological protections in vaccinated pigs, however, failed to observe the complete virological protection in vaccinated pigs. Thus, it indicates that the vaccine does not provide complete protection or sterile immunity.
The author responds: Yes, thank you. You have made an accurate point. The authors use this phrase because sterilizing immunity is an expression of an immune response that, while not perfect, prevents the vaccine virus from replicating into the host (pig). In addition, in our previous ASF experiments, we performed post-vaccine viral clearance experiments, and some individuals did not achieve complete viral clearance (unpublished data). However, no infections occurred in cohabiting livestock. In conclusion, the reason why we cited (interpreted) Reference No. 16 & 19 was to confirm replication inhibition of the challenge virus after vaccination. Although not confirmed by marketing age, the results for rectal and oral attached to the reviewer's 6th virus shedding question show that low level genome copies are confirmed in oral swabs in 2 out of 4 dogs on day 56 when vaccinated alone (Table 2).
Additional author comments: We also do not believe there is a perfect live attenuated vaccine. Viral clearance after vaccination refers to the process by which a vaccine helps the immune system clear the virus from the body. An effective vaccine will stimulate an immune response that will reduce the amount of virus and ultimately eliminate the virus. However, the speed and efficiency of viral clearance can vary depending on the type of vaccine, the health status of the animal, and the specific virus. For example, research on ASF is still analyzing the effect of vaccination on reducing the presence of the virus in the body.
Comments 6. Lines 384-390: The authors are aware of the importance of virus shedding in vaccinated pigs. However, no data is presented in this study. Please include virus shedding data.
The author responds: Thank you for your precise point. Yes, we have conducted viral shedding experiments using qPCR method on oral and rectal swabs. The reason why we did not include it in the current manuscript is that we focused on whole blood results and vaccine efficacy, and if we added data on oral and rectal, the volume of the manuscript would be too large. However, we have attached the results of qPCR (Ct<45) for oral and rectal swabs in this experiment as Table 1-2. We will act according to the reviewer's decision after checking.
Summary) Vaccine genome shedding was identified at low levels in 2 out of 4 animals per individual, with somewhat more frequent shedding in the oral than the rectal. In the post-vaccine challenge, shedding was relatively consistent across all groups up to the time of euthanasia.
Summary) Vaccine genome shedding was detectable in all but 1 pig from day 14 post-vaccination and was mostly detected in oral swabs by day 48 but was detected at very low levels in the oral cavity of 2 of 4 pigs by day 56.
Comments 7. One of the key findings in this study is the persistent infection of the
vaccine strain in pigs up to 56 DPV. It could provide the favorable environments
for viruses to acquire genetic mutations. Thus, it is important to determine the
viral sequences in vaccinated pigs. In particular, viremia in vaccinated pigs on
the day of the challenge may result in the recombination event between the
vaccine strain and the challenge strain because recombination contributes to the
genetic expansion of ASFV and the reduction in the efficacy of the current
vaccine in the field. Please perform the NGS on the samples from Group A and
B to support the safety of vaccine.
The author responds: Thank you for pointing out the key points. Our researchers are working with government agencies and universities to collect whole blood after vaccination and send it to commercial (Celemics) organizations for NGS analysis. We are currently waiting for the NGS results. As a matter of fact, we plan to analyze the results once they are available and submit another manuscript to Animals. However, referring to the first point made by the reviewer about the PIPEC cell line NGS analysis, although the analysis is ongoing, initial analysis indicates no unusual genetic changes in P19, a serial passage from the vaccine strain initially introduced by USDA.
Reviewer 3 Report
Comments and Suggestions for Authors
The authors reported thath clinical trials of ASFV-G-ΔI177L/ΔLVR vaccine candidates have been markedly limited. In this study, they strictly followed the WOAH Biological Standard Commission vaccine production standards and conducted clinical trials with a high-purity vaccine. Vaccine purity was achieved by thoroughly eliminating the possibility of mutations during cell line passage. They report excellent efficacy and safety in homologous challenge after low-dose ASFV-G-ΔI177L/ΔLVR vaccination, also demostratited ASF vaccine antibodies were maintained at a substantially high level for approximately 2 months after vaccination. These results confirm that the ASFV-G-ΔI177L/ΔLVR vaccine candidate can play a role as a relief pitcher for ASF infection in the swine industry.
I just suggest
-to mentionate more vaccine candidates i.e. recombinant ASFV P30 protein and production of monoclonal antibodies (Liberti et al.).
-to explore possible perspectives, adding more investigations such as citokine profiling during the vaccination trials.
Author Response
Reviewer 3.
Comments and Suggestions for Authors
Comments 1: The authors reported that clinical trials of ASFV-G-ΔI177L/ΔLVR vaccine candidates have been markedly limited. In this study, they strictly followed the WOAH Biological Standard Commission vaccine production standards and conducted clinical trials with a high-purity vaccine. Vaccine purity was achieved by thoroughly eliminating the possibility of mutations during cell line passage. They report excellent efficacy and safety in homologous challenge after low-dose ASFV-G-ΔI177L/ΔLVR vaccination, also demonstrated ASF vaccine antibodies were maintained at a substantially high level for approximately 2 months after vaccination. These results confirm that the ASFV-G-ΔI177L/ΔLVR vaccine candidate can play a role as a relief pitcher for ASF infection in the swine industry.
I just suggest
-to mention more vaccine candidates i.e. recombinant ASFV P30 protein and production of monoclonal antibodies (Liberti et al.).
-to explore possible perspectives, adding more investigations such as cytokine profiling during the vaccination trials.
The author responds: Thank you very much for your understanding of the core of the paper. I will refer to the ASFV P30 protein and production of monoclonal antibodies (Liberti et al.) mentioned by the reviewer and use it well in my next research. In addition, I will add research on cytokines for each group after ASFV-G-ΔI177L/ΔLVR vaccination through additional research soon. Once again, thank you for your valuable suggestions.
Round 2
Reviewer 1 Report
Comments and Suggestions for Authors
Following the previous round of review and subsequent revisions by the authors, the manuscript's quality has markedly improved.
Author Response
Reviewer 1:
Comments and Suggestions for Authors
Comments 1: In this study, Choi et al. aimed to evaluate the efficacy, safety, and long-term persistence of antibodies elicited by the ASFV-G-ΔI177L/ΔLVR vaccine. The findings confirmed the vaccine's efficacy and safety, demonstrating that antibody levels remained stable for approximately two months. The safety and efficacy of the ASFV-G-ΔI177L/ΔLVR vaccine candidate strain, along with the induced antibody response, have already been investigated. Consequently, the significance of this research is somewhat constrained. Some issues need to be addressed by the authors.
- The group A lacks a high-dose immune group to evaluate the safety of ASFV-G-ΔI177L/ΔLVR.
The author respond: Thank you for your key questions. Our clinical investigators did not include the results of high-dose vaccine inoculation in this paper for the following reasons: We have conducted more than 10 experiments under ABSL-3 conditions for all of our ASF vaccine trials. High-dose vaccine inoculation is not in this experiment, but in previous experiments, we have already verified the efficacy and safety of the vaccine (100% survived) using 5 animals in each group with ASFV-G-ΔI177L/ΔLVR, ASFV-G-ΔI177L/ΔLVR, 106TCID50/dose. However, as suggested by the reviewer, we have organized the results of vaccine genome copy (qPCR) and vaccine antibodies (cELISA) into Table 1 (Genome copy) and Table 2 (Vaccine antibody) for easier understanding.
Table 1. Vaccine genome copies after vaccination (ASFV-G-ΔI177L/ΔLVR).
Table 2. Vaccine antibody after vaccination.
Comments 2. The safety of the live attenuated ASF vaccine should be assessed by monitoring virus shedding; however, this study did not perform these tests.
The author responds: Thank you for pointing out this important aspect.
The reason why we did not include it in the current paper is that we focused on whole blood results and vaccine efficacy, and if we added data on oral and rectal, the volume of the paper would be too large as well as the content, so we could not attach it. The qPCR (Ct<45) results for Rectal and Oral swabs are attached as Table 1-2. Although not confirmed by marketing age, the results for rectal and oral attached to the reviewer's virus shedding question show that low level genome copies are confirmed in oral swabs in 2 out of 4 dogs on day 56 (dpv) when vaccinated alone (Table 2). After checking, we will decide whether to attach them to this paper according to the reviewer's decision.

Reviewer 2 Report
Comments and Suggestions for Authors
The points have been addressed well in the rebuttal letter, however, they are not reflected in the main manuscript. The authors may use the supplementary materials if there is any concern on the volume of the main manuscript. Please include all additional information in the main manuscript.
Comment 1: Please provide the P9 and P19 sequences, including sequencing depth. Also, please deposit the raw data in the public resource.
Comment 2: Please add method of WGS and virus titration in the main manuscript.
Comment 3: The authors addressed the critical points in the authors responds to comment 5. Please add them in the main manuscript.
Comment 4: There is a plenty of empty space in the main manuscript. It can accommodate the critical data, such as virus shedding, and the discussion on these data.
Author Response
2nd Reviewer’s Respond:
Comments and Suggestions for Authors:
The points have been addressed well in the rebuttal letter, however, they are not reflected in the main manuscript. The authors may use the supplementary materials if there is any concern on the volume of the main manuscript. Please include all additional information in the main manuscript.
Comment 1: Please provide the P9 and P19 sequences, including sequencing depth. Also, please deposit the raw data in the public resource.
The Author Respond 1: Thank you for your important point. We will ask the reviewer about the issue of including the P9 and P19 sequences (NGS) raw data in the paper. Of course, we will send the raw data to the reviewer (see attachment of NGS-P9_sequence & NGS-P19_sequence). The author's concern is that we haven't even finished analyzing the data yet, can we insert the P9 & P19 raw data into the supplementary materials? If it is possible, we will attach them separately as supplementary in the paper.
Comment 2: Please add the method of WGS and virus titration in the main manuscript.
The Author Respond 2: As mentioned in Response 1, we did not describe the Method of WGS in the Material and Method section because we are doing NGS analysis for P9 and P19 sequences. However, we described the PIPEC culture (Please see Line 92-100) and Virus titration (Please see Line 104-109) with citations.
Comment 3: The authors addressed the critical points in the authors responds to comment 5. Please add them in the main manuscript.
The Author Respond 3: Thank you for your suggestion for the Discussion section as well. We have added your suggested "critical points" section to the main manuscript (Please see Lines 413-421).
Comment 4: There is a plenty of empty space in the main manuscript. It can accommodate the critical data, such as virus shedding, and the discussion on these data.
The Author Respond 4: We appreciate the reviewer's detailed analysis of the paper and his efforts to improve the quality of "Animals" Journal. We have attached the results of vaccine virus shedding in Tables 1 & 2 as Table Supplementary1: Group A, Detection of vaccine genome copies by qPCR in Rectal and Oral swab samples from the challenge after ASFV-G-ΔI177L/ΔLVR vaccination & Table Supplementary 2: Group B (Long-term), Vaccine genome copy using qPCR in Rectal and Oral swab samples after ASFV-G-ΔI177L/ΔLVR vaccination. We have added the experimental methods and results of virus shedding (Please see Lines 246-263) as well as the Discussion section (Please see Lines 400-406 & 441-446).
Round 3
Reviewer 2 Report
Comments and Suggestions for Authors
The authors mentioned the importance of the genetic stability during passages for production (lines 85-87) and selection of P19 as MSC after confirming various modifications (lines 99-101). However, the authors responded the NGS analysis has not been completed in the rebuttal letter. These statements contradicts each other. Please provide the correct information and data to readers.
Author Response
Reviewer 2 (3rd Round):
Comments and Suggestions for Authors
The authors mentioned the importance of the genetic stability during passages for production (lines 85-87) and selection of P19 as MSC after confirming various modifications (lines 99-101). However, the authors responded the NGS analysis has not been completed in the rebuttal letter. These statements contradicts each other. Please provide the correct information and data to readers.
The author answers 1: Once again, thank you for the reviewer's thoughtful point. In response to the reviewer's comment, we requested the results for NGS and fortunately we have received the draft NGS analysis for P9 and P19 (Please see attached file). We will present the results as follows (also find Excel file: NGS P9 and P19-2025). In addition, we have reviewed the WOAH, Biological Standards Commission, Annex 16. Item 5.1 Chapter 3.9.1. African swine fever. Guideline states that "MSV must be confirmed using appropriate methods (e.g. such as NGS)." However, since it is not possible to perform NGS analysis every 5 passages, our choice of appropriate method is the end-point dilution assay (Arch. Virol. 1975, 49, 323-328), which confirms by PCR the absence or presence of the MGF505 7R-9R fusion during the PIPEC passage of the vaccine strain. The vaccine strain without the mutation was confirmed by PCR and is being managed for animal immunization after measuring the vaccine titer. We have been managing the vaccine strain in such a detailed manner, which is why we have expressed it in lines 99-101.
※Note: In fact, the NGS analysis is not limited to PIPEC passages, but also whole blood from vaccinated pigs is being analyzed to determine the presence of variation between vaccination intervals. We are planning to submit another manuscript with the results of all these NGS analyses. The tentative title of the paper is "Characterization of the vaccine genome in PIPEC serial passages of ASFV-G-ΔI177L/ΔLVR and in whole blood of inoculated animals using NGS analysis."
Therefore, the efficacy and safety of ASF vaccine candidate focusing on vaccine genome copy and vaccine antibody (including long-term maintenance) are the key contents of this paper (Animals). Your understanding of this will be greatly appreciated. The results of NGS analysis of vaccine candidate PIPEC passage and whole blood after vaccination of pigs will be submitted in the next paper.
